# Genotypic Characterization of Clinical Isolates of *Staphylococcus aureus* from Pakistan

**DOI:** 10.3390/pathogens10080918

**Published:** 2021-07-21

**Authors:** Saeed Khan, Bernard S. Marasa, Kidon Sung, Mohamed Nawaz

**Affiliations:** Division of Microbiology, National Center for Toxicological Research, US Food and Drug Administration, Jefferson, AR 72079, USA; bernard.marasa@fda.hhs.gov (B.S.M.); kidon.sung@fda.hhs.gov (K.S.); Mohamed.nawaz@fda.hhs.gov (M.N.)

**Keywords:** methicillin resistance, virulence, genes, MLST, SCC*mec*, *Staphylococcus aureus*, evolution

## Abstract

In this study, we compared pulsed-field gel electrophoretic (PFGE), multilocus sequence typing (MLST), Staphylococcal cassette chromosome *mec* (SCC*mec*), *spa* typing, and virulence gene profiles of 19 Panton–Valentine leucocidin (PVL)-positive, multidrug-, and methicillin-resistant clinical *Staphylococcus aureus* (MRSA) isolates obtained from a hospital intensive care unit in Pakistan. The isolates exhibited 10 pulsotypes, contained eight adhesin genes (*bbp, clfA, clfB, cna, fnbA, fnbB, map-eap,* and *spa*), 10 toxin genes (*hla, hlb, hld, hlg, pvl, sed, see, seg, seh,* and *tst*), and two other virulence genes (*cfb, v8*) that were commonly present in all isolates. The *spa*-typing indicated seven known *spa* types (t030, t064, t138, t314, t987, t1509, and t5414) and three novel *spa* types. MLST analysis indicated eight ST types (ST8, ST15, ST30, ST239, ST291, ST503, ST772, and ST1413). All isolates belonged to the *agr* group 1. Most of the isolates possessed SCC*mec* type III, but some isolates had it in combination with types SCC*mec* IV and V. The presence of multidrug-resistant MRSA isolates in Pakistan indicates poor hygienic conditions, overuse of antibiotics, and a lack of rational antibiotic therapy that have led to the evolution and development of hypervirulent MRSA clones. The study warrants development of a robust epidemiological screening program and adoption of effective measures to stop their spread in hospitals and the community.

## 1. Introduction

MRSA is the major source of hospital-acquired infections and is of particular concern due to its involvement in high incidences of morbidity and mortality worldwide [1,2,3]. Resistance to methicillin is conferred by a *mecA* gene that was first discovered in 1961 among nosocomial *S. aureus* isolates, and since then, it has been independently transferred multiple times into the *S. aureus* chromosome rather than originating from a single ancestral strain [4]. While the incidence of MRSA infections and their epidemiology is well documented in western countries, data from the National Nosocomial Infections Surveillance System of Pakistan suggests that incidences of MRSA infections have increased from 35.9% to 66.7% in Pakistan during 2009–2019 [5,6,7]. The presence of other virulence factors, extracellular enterotoxin genes, and Panton–Valentine leucocidin cytotoxin (PVL) genes makes MRSA highly pathogenic and difficult to treat [8]. This organism causes a variety of infections, such as boils, furuncles, styes, impetigo, and other superficial infections, in humans [9,10]. It is also known to cause serious infections, such as pneumonia, deep abscesses, osteomyelitis, endocarditis, phlebitis, mastitis, and meningitis, in immunocompromised and severely ill individuals [11,12]. Among life-threatening diseases, MRSA has been implicated in unusually invasive pathogenic diseases, such as severe septicemia, necrotizing fasciitis, and pneumonia. Hospital-acquired MRSA (HA-MRSA) strains also exhibit resistance to multiple antibiotics, including macrolides and fluoroquinolones, which poses serious challenges to the treatment of infections [13,14]. MRSA is no longer limited to hospital settings; community-acquired MRSA (CA-MRSA) is also becoming a major threat to public health and can be transmitted in homes, workplaces, and child-care facilities [15]. The association of Panton–Valentine leucocidin (PVL) genes in CA-MRSA causes infections in healthy young and immunocompetent hosts, sometimes with fatal outcomes [16]. PVL genes cause leukocyte destruction and necrotizing pneumonia that can kill patients within three days. Community-acquired pneumonia (CAP) afflicts young and healthy individuals and leads to more than 40% mortality [17]. PVL-positive MRSA has played a critical role in several outbreaks and causes life-threatening bacterial infections. PVL-positive CA-MRSA strains usually cause soft-tissue infections, but PVL has also been viewed as a key virulence factor that helps bacteria in targeting and killing neutrophils. In one of the studies, however, this view was challenged, wherein the removal of PVL from CA-MRSA strains did not result in a loss of infectivity or neutrophils destruction [18]. It has been shown that the worldwide spread of PVL-positive CA-MRSA was not due to a single clone, but rather, it emerged multiple times on different continents [19]. PVL has been suspected in increasing the expression of staphylococcal protein A, which is an important pro-inflammatory factor for pneumonia. The presence of PVL has also been shown to increase virulence in some strains of *S. aureus*. Most CA-MRSA isolates studied have been shown to harbor PVL genes [20].

The genome of *S. aureus* contains large areas of differences, even within species, and some of them carry virulence factors, toxins, and the genes for antimicrobial resistance [21]. It is believed that the *S. aureus* genome contains ~22% of dispensable genetic material, as well as several dispensable regions of its genome, and these genetic variations are likely responsible for a poorly understood phenomenon of disease and host specificity. Several techniques, such as molecular typing of *S. aureus* using amplified fragment length polymorphism [22], pulsed field gel electrophoretic analysis (PFGE) [23], and the presence of virulence genes and factors [24], have been used successfully to determine clonal relationships and for source tracking in epidemiological screening of these isolates. In this study, we used some of these techniques to explore the diversity and distribution of prevalent clonal isolates, including the presence of virulence factors and toxin genes, PFGE, ST, *spa* typing, and SCC*mec* patterns, in six PVL-positive CA-MRSA and thirteen PVL-positive HA-MRSA isolates from Pakistan. The data yielded novel information that could be helpful in understanding the evolution and emergence of PVL-positive CA-MRSA isolates, risk assessment of the threats posed by these isolates, and designing strategies to manage and control *S. aureus* infections.

## 2. Results

### 2.1. Antimicrobial Susceptibility and Resistance Genes

Disk diffusion assays indicated that all isolates were resistant to multiple antibiotics, including ampicillin, chloramphenicol (except for isolate 50), erythromycin (except for isolate 15), gentamicin, kanamycin, oxacillin, methicillin, penicillin, and tetracycline, and all were susceptible to vancomycin. The antibiograms and MIC data are presented in Appendix A. PCR amplification of the resistance genes that confer resistance to chloramphenicol, erythromycin, tetracycline, trimethoprim, methicillin, and β-lactams indicated the presence of multiple genes that conferred resistance to these antibiotics. Most of the isolates resistant to chloramphenicol exhibited the presence of chloramphenicol resistance genes, *cat(pC194), cat(pC221)*, and *cat(pC223)*, except for isolate 10, which lacked *cat(pC194)*, and isolate 50, which was susceptible to chloramphenicol and did not possess any of the genes mentioned above (Table 1, Figure 1C). A majority of the isolates resistant to erythromycin contained the *ermB* gene alone or in combination with the *sat4* gene. Isolate 15, which was susceptible to erythromycin, did not contain any of these genes. Among the tetracycline resistance genes tested, most isolates showed *tetM, tetS,* and *tetW* genes, except for isolates 50 and 52 that lacked the *tetM* gene. Isolates 32 and 33 contained an additional *tetK* gene, and isolates 10, 25, and 32 contained the *tetL* gene. All isolates contained the trimethoprim resistance gene *dfrA* and the methicillin resistance gene *mecA*. Of the 19 isolates, all but five isolates (isolates 10, 15, 25, 30, and 37) contained the *blaZ* gene.

### 2.2. PFGE Analysis

Separation of the *Sma*I-digested chromosomal DNA fragments from the isolates by PFGE indicated 10 different patterns. Nineteen MRSA isolates were grouped into 10 pulsed-field gel electrophoretic (PFGE) groups. PFGE profile 1 (Figure 2, Table 2) was comprised of six MRSA isolates of perirectal origin (isolates 32, 33, 34, 38, 41, 48). PFGE profile 2 consisted of five isolates of nasal origin (isolates 15, 25, 31, 37, 40). PFGE profiles 3 to 10 were represented by single isolates 10, 30, 35, 42, 49 and 50, 51, and 52, respectively (Figure 2, Table 2).

### 2.3. PVL Gene Amplification and MLST and spa-Typing

Sequencing of ~450-bp internal fragments of seven housekeeping genes followed by mapping via the https://pubmlst.org/organisms/staphylococcus-aureus/ (accessed on 20 April 2021) indicated eight different MLST patterns (Table 2). The most common pattern exhibited by 9 of 19 isolates was ST239 with the allelic profile 2-3-1-1-4-4-3. Three isolates presented the ST8 profile (3-3-1-1-4-4-3), two isolates showed ST772 (1-1-1-1-22-1-11), and one of each of the five other isolates had ST30 (2-2-2-2-6-3-2), ST503 (2-2-22-13-3-2), ST291 (3-37-19-2-20-26-32), ST15 (13-13-1-1-12-11-13), and ST1413 (6-5-6-2-162-14-5) profiles. Sequence analysis of the X region of the *spa* gene indicated eight different *spa*-types, the most common being the *spa*-type t030 represented by seven isolates (31, 32, 33, 34, 35, 41, and 48) (Table 2). The *spa*-type t064 was present in isolates 25 and 37; these isolates were also found to have two nucleotide substitutions in the 3′ flanking region (Table 3). Sequence analysis revealed *spa*-types t138 in isolates 40 and 50 and t5414 in isolates 42 and 49. *Spa*-types t314, t987, t1149, and t1509 were present in isolates 10, 30, 51, and 52, respectively. Three new *spa*-types were identified in isolate numbers 15, 38, and 51. Each new *spa*-type differed from each other based on flanking and variable X region sequences (Table 3). Isolate 15 presented two new 24-bp VNTRs and different 5′ and 3′ flanking sequences, and isolate 38 showed a new 24-bp VNTR immediately before the 3′ flanking sequence. Although isolate 51 exhibited a known *spa* type, it contained a novel 5′ flanking sequence. The results of PCR assays for different leukocidin genes are shown in Table 4.

### 2.4. SCCmec and agr Group Typing

Of the 19 MRSA isolates tested for SCC*mec* types, 10 isolates showed SCC*mec* type III (isolates 15, 30, 32, 33, 34, 35, 41, 42, 48, 49), one showed SCC*mec* III + IVa (isolate 31), two presented SCC*mec* III + V (isolates 10, 51), two exhibited SCC*mec* III + IVa + V (isolates 50, 52), three showed SCC*mec* IVa (isolates 25, 37, 40), and one presented SCC*mec* V (isolate 38). All detected SCC*mec* types belonged to *agr* group I (Table 2).

### 2.5. Capsular Polysaccharides

Of the 19 MRSA isolates tested, seven isolates (25, 37, 38, 42, 49, 50, 52) showed cap 5, and 10 isolates (10, 15, 30, 31, 32, 33, 35, 40, 41, 48) exhibited cap 8. Two of the isolates (isolates 34 and 51) were non-typable (Table 2).

### 2.6. Toxin, Adhesin, Virulence and Hemolysin Genes

Of the 12 adhesin genes (*fnbA, fnbB, clfA, clfB, cna, spa, sdrC, sdrD, sdrE, bbp, ebpS,* and *map-eap*) tested, eight genes (*bbp, cna, clfA, clfB, fnbA, fnbB, map-eap,* and *spa*) were common to all isolates (Figure 1A,B, Table 4). PCR assays of the 19 toxin genes (*edin*, *eta, etb, hla, hlb, hld, hlg, hlg2, pvl, sea, seb, sec, sed, see, seg, seh, sei, sej, tst*) indicated the presence of 10 genes (*hla, hlb, hld, hlg, pvl, sed, see, seg, seh, tst*) that were common to all isolates (Figure 1A,B, Table 4). The two PVL genes, *lukS-PV* and *lukF-PV*, yielded PCR products at 538 and 460 bp, respectively. They were present in most of the isolates tested but absent from isolates 15, 25, 31, and 37 (Table 4). The *lukM* gene, which acts like *lukF-PV* in combination with *lukS* or a hemolysin gene *hlg2*, was also absent from these isolates. Five other virulence genes (*arcA, cfb, chp, ica, v8*) were tested, and only two (*cfb, v8*) were common to all isolates. Based on the hemolysin assays on sheep blood agar plates and PCR analysis, the isolates were found to possess α, β, or γ hemolysin genes. The hemolysin gene α was present in isolates 15, 25, 37, 42, 48, and 52 (Table 4). Those that contained hemolysin gene β were isolates 38, 40, 49, 50, and 51. Isolates 10, 30, 31, 32, 33, and 34 exhibited the γ hemolysin gene (Table 4).

## 3. Discussion

MRSA isolates pose a serious threat to public health and are a global cause of concern regarding health care- and community-associated infections [25,26]. Studies of invasive MRSA isolates have recognized multiple clones of MRSA that are circulating worldwide, where strains of clonal complex (CC) 5 and CC8 are the most ubiquitous and most diverse [27,28]. Population-based studies of invasive MRSA infections in the United States indicate that MRSA clones of ST5 (CC5) and ST8 (CC8) are the predominant causative strains in health care-associated bacteremia [29]. There are relatively few studies of the distribution of MRSA and their prevalence among Pakistani isolates, which make it difficult to compare MRSA strain types in Pakistan to those in other countries, such as the most recent European MRSA surveillance data and the US data [30,31]. Only a few studies from Pakistan characterized MRSA isolates for the presence of PVL genes and SCC*mec*, PFGE, and ST typing [30,31,32]; none of these studies explored the presence of virulence, adhesion, and enterotoxin genes among MRSA isolates from this geographical region. We used PFGE, MLST, *spa*, and SCC*mec* typing techniques to understand the genotypic diversity and relationship, evolutionary changes, distribution pattern of VNTRs within the *spa* region, SCC*mec*, *agr* grouping, and distribution of various virulence, toxin, and antimicrobial resistance genes among 19 CA- and HA-MRSA isolates from Pakistan. The data provided a wealth of new information, and the 19 isolates exhibited 10 different PFGE profiles and eight different ST patterns. All isolates with identical PFGE profiles did not exhibit the same ST profile, and likewise, isolates with different PFGE profiles shared similar ST profiles. For example, isolate 38 from the PFGE group I, which contained six isolates, exhibited the ST8 profile, while others in the group exhibited the ST239 pattern. Moreover, five MRSA isolates representing PFGE profile 2 showed three different ST profiles (ST239, ST8, and ST30). ST239 and ST8 profiles in this group matched those seen in PFGE group I. It has been reported that ST239 types evolved as a single locus variant of ST8 and ST30 [33,34]. Here, it appears that the ST239 observed in isolates with PFGE profiles 1, 2, 4, and 5 evolved either from ST8 or ST30 clones that once were predominantly present in the MRSA isolates from Pakistan, according to region-specific information from the MLST database (https://pubmlst.org/organisms/staphylococcus-aureus/, accessed on 20 April 2021). Our study results agree with earlier reports of the prevalence of ST239 and ST8 clones among MRSA isolates from Pakistan [31,32]. MRSA isolates exhibiting independent PFGE profiles 3, 6, 7, 8, 9, and 10 represented by isolates 10, 42, 49, 50, 51, and 52, respectively, appear to have evolved independently. These isolates presented five known ST types (ST1413, ST772, ST503, ST291, and ST15). Among these isolates, CA-MRSA isolates 42 and 49 showed a common ST772 profile and approximately 80% similarity based on their PFGE profiles. Isolate 42 seemed to have either evolved from isolate 49 due to deletion or isolate 49 evolved from isolate 42 due to acquisition of extra genetic elements.

CA- and HA-MRSA isolates are often differentiated by the type of SCC*mec* elements and drug-resistant phenotypes present on a 21- to 67-kb mobile genetic element [35,36]. These elements have been classified into five major types and subtypes. Types IV and V are present on smaller DNA fragments and are easily transferable to other staphylococcal isolates compared to types I, II, and III, which are present on larger DNA fragments. A majority of HA-MRSA isolates are reported to carry longer transposable genetic elements, such as SCC*mec* type I, II, and III, and multidrug resistance genes, whereas CA-MRSA isolates are believed to carry a smaller number of genes, such as SCC*mec* type IV and V, and are not multidrug-resistant [37]. In our study, we noticed that five of the six CA-MRSA isolates were resistant to multiple drugs, such as chloramphenicol, erythromycin, tetracycline, trimethoprim, oxacillin, and methicillin, and carried SCC*mec* III. On the other hand, HA-MRSA isolates, apart from being resistant to the above antibiotics, also carried SCC*mec* III alone or in combination with SCC*mec* IVa and SCC*mec* V. The combined SCC*mec* III + IVa or SCC*mec* III + V types indicate a recombination between CA- and HA-MRSA strains. Few reports of combined SCC*mec* types are available [31,38] and indicate a recombination between CA- and HA-MRSA isolates. A study of CA- and HA-MRSA isolates from Pakistan indicated that a majority of the HA-MRSA isolates were either SCC*mec* type III or its variants, and a majority of the CA-MRSA were SCC*mec* type IV or its variants [38,39]. A recently published study [40] from this region indicated that 47% of the clinical MRSA strains carried SCC*mec* type III and were PVL positive, while 29% exhibited SCC*mec* type IV and were PVL negative. In our study, the type distribution was as follows: SCC*mec* type III (53%), type IVa (16%), type V (5%), types [III + IVa] (10%), types [III + V] (10.5%), and types [III + IVa + V] (10.5%). Data from our study and one published earlier [39] clearly indicate that the SCC*mec* type III is predominant in this geographical region; the second most prevalent is SCC*mec* type IV. Moreover, the earlier study indicated that only PVL-positive isolates showed SCC*mec* type IV elements, and those with the SCC*mec* type III elements were PVL negative. However, a closer look at the data from that study indicated that an almost equal number of strains with SCC*mec* type III also exhibited the presence of PVL genes. In our study, we did not see any correlation between CA-MRSA, SCC*mec* type, and the presence of PVL genes, as was emphasized in the earlier study [39]; PVL genes were present in all isolates included in our study. The presence of PVL genes in combination with the *tst* gene that produces toxic shock syndrome toxin-1 (TSST-1) and causes toxic shock syndrome is reportedly a rare occurrence [41], but all of the isolates examined in this study contained both the PVL genes and the *tst* gene, which may indicate hypervirulence.

HA-MRSA isolates are known to cause serious and life-threatening diseases, while CA-MRSA isolates are believed to cause soft-tissue infections [42]. These concepts are now changing with the evolving nature of CA-MRSA as it is becoming a serious threat to public health due to its potential to cause necrotic lesions in skin or mucosa and necrotic hemorrhagic pneumonia, often with fatal outcomes [42]. Of the 19 isolates used in this study, six patients acquired MRSA in a community setting, and 13 acquired MRSA in a hospital setting. Two of the four CA-MRSA and four of the 13 HA-MRSA patients died while undergoing prolonged treatment. One CA-MRSA patient died due to septic shock, and one other CA-MRSA patient died due to chronic renal failure. Among four HA-MRSA patients that died, one died due to complications from renal failure, one from respiratory failure due to kyphoscoliosis, one from tetanus/lockjaw, and one from Guillain–Barre syndrome (GBS). Among the *S. aureus* isolates from the six patients that died, one exhibited the SCC*mec* type III (CA-MRSA isolate 33), two showed IVa (CA-MRSA isolate 25 and HA-MRSA isolate 40), one presented SCC*mec* [III + IVa] (HA-MRSA isolate 31), and two indicated SCC*mec* [III + V] (HA-MRSA isolates 10 and 51). Interestingly, MRSA isolates carrying SCC*mec* IVa or a combination of SCC*mec* [III + IVa] and SCC*mec* [III + V] were apparently associated with a higher number of deaths than those that carried SCC*mec* III. MRSA isolates from nine of the 13 patients who survived infection showed SCC*mec* III, one presented IVa, one showed V, and two contained a combination of SCC*mec* [III + IVa +V]. It appears that the combination of SCC*mec* III with either IVa or V contributes to the highest level of virulence. However, the patient infected with MRSA isolate 50, and carrying SCC*mec* [III + IVa + V] elements, did not exhibit life-threatening conditions. Recovery of this patient suggested that other factors, such as timely intervention, treatment options, and patient’s condition when originally admitted, could play a crucial and effective role in combating these life-threatening infections.

The first step in the process of bacterial infection is the adherence of bacteria to human epithelial cells; this property has been used to define the pathogenicity of an infecting agent. Fibronectin binding proteins and their corresponding *fnbA* and *fnbB* genes have been proposed to be some of the major ligands on the staphylococcal cell surface that help *S. aureus* adhere to epithelial cells [43]. Approximately 90% of clinical isolates have been shown to contain the *fnbA* gene, and only 20% have been shown to harbor the *fnbB* gene [43]. The presence of both genes in *S. aureus* provides strong adherence properties and heightens pathogenicity. In our study, we observed that all MRSA isolates contained both *fnbA* and *fnbB* genes. In addition, eight of the 12 adhesion genes were commonly present in all isolates which potentially allows the bacteria to stick to a variety of surfaces and form a biofilm that renders them impermeable and resistant to various antimicrobial agents. One of the adhesion genes, namely *ebpS*, was predominantly present in (6/8) isolates of perirectal origin rather than in nasal (2/11) isolates; implications of this difference are not currently known.

While the presence of adhesion genes offers *S. aureus* isolates the ability to adhere to different surfaces, the presence of hemolysin and toxin genes allow them to be highly pathogenic. Most isolates from human, bovine, and food sources have been reported to primarily possess the α hemolysin gene *hla* [44], which causes incomplete or partial hemolysis, but in our study, we noticed the presence of α, β, and γ hemolysin genes in both CA- and HA-MRSA isolates. These isolates were also found to contain several superantigen genes (*sea, seb, sec, sed, see, seg, seh, sei, sej, tst*) that have been reported to be responsible for the production of pro-inflammatory cytokines, infective endocarditis, scarlet fever, and possibly septic shock syndrome [45,46,47]. Several of these genes have been identified and reported in over 70% of *S. aureus* isolates [48]. Among the staphylococcal strains, *tst* is the most common superantigen gene associated with staphylococcal toxic shock syndrome; a high prevalence of *sea* [45,46] and the superantigen genes *seg* and *sei* [47] have also been shown in patients with *S. aureus*–mediated septic shock and scarlet fever. The presence of these genes among *S. aureus* isolates evaluated in this study indicates a possibility that these isolates also have the potential to cause toxic shock syndrome. However, most patients admitted to intensive care units exhibited a variety of symptoms, and only one patient infected with CA-MRSA isolate 33 presented with this syndrome. This finding implies that other factors, such as differences in gene expression levels, biological activity, interactions with other cellular components, and genetic differences among individuals, may also play an important role in the outcome of *S. aureus* diseases and their manifestation, rather than the mere presence of such genes in associated isolates. The “*agr* locus”, an accessory gene regulator, is a two-component quorum-sensing signaling pathway that controls the expression of many virulence factors in *S. aureus*. It encodes an autoinducing peptide (AIP), which exhibits the amino acid polymorphism in its sequence, and based on the polymorphism of the AIPs and their corresponding receptors, four major *agr* groups (I to IV) have been assigned [49]. The AIPs belonging to different groups are usually mutually inhibitory, but those within a group can activate the *agr* response in other member strains. All isolates used in this study belonged to the *agr* group I.

In summary, the present study identified eight ST types, three novel *spa* types, a combination of unique SCC*mec* types, specific virulence and pathogenicity markers, and their distribution and prevalence in CA- and HA-MRSA isolates from Pakistan. The most common *spa* type was t030, and the most prevalent ST type was ST239. All the isolates belonged to the *agr* group 1. Most of the isolates possessed SCC*mec* type III, and it was found to be predominantly present in these isolates either as a single SCC*mec* type or in combination with SCC*mec* IV and V. The presence of identical PFGE, MLST, and SCC*mec* patterns, as well as virulence and toxin genes in CA- and HA-MRSA isolates, suggests a transmission of these isolates and their traits between CA- and HA-MRSA and an evolutionary trend of *S. aureus* isolates in this geographical region. Our findings also suggest an erosion of a fine line of difference between CA-and HA-MRSA isolates, establishment of CA-MRSA isolates in hospital settings, and their evolution into powerful pathogens, all posing a serious challenge to public health both in hospital and in community settings. Poor hygienic conditions and the flow of people between hospital and community settings has provided ample opportunity for these pathogens to recombine and pass on traits to other organisms. These study findings clearly warrant a need for continued monitoring, epidemiological screening, and implementation of extreme measures to contain and control the spread of such pathogens.

## 4. Materials and Methods

### 4.1. Isolation, Identification, and Characterization of Isolates

Several presumptively positive *Staphylococcus* spp. isolates of the nasal and perirectal origin were obtained from multiple patients admitted to the intensive care unit (ICU) of a tertiary health care facility in Rawalpindi, Pakistan. Each of the strains used in the study was isolated from individual patients. Nineteen of these isolates were identified as methicillin-resistant *Staphylococcus aureus* based on their biochemical characterization with the Vitek identification method, *nucA* expression*,* and coagulase profile. HA-MRSA isolates represented the isolates from patients that were undergoing treatment in the hospital for something else but acquired MRSA infection during hospital stay. CA-MRSA isolates were obtained from the individuals that came to the hospital with MRSA infection and were admitted. Since methicillin-resistant strains cause severe and sometimes fatal infections, we used these strains for detailed molecular characterization.

### 4.2. Antibiogram and MIC Using a Broth Dilution Method

All isolates were tested for antimicrobial susceptibility against the criteria of the Clinical and Laboratory Standards Institute (CLSI) [50,51]. The following antibiotics, with their concentrations noted after their names in µg mL^−1^, were tested to determine the susceptibility of *S. aureus* isolates: ampicillin-10, chloramphenicol-30, erythromycin-15, gentamicin-120, kanamycin-30, levofloxacin-2, oxacillin-1, methicillin-5, norfloxacin-5, penicillin-10, tetracycline-30, and vancomycin-30 (AB BIODISK, Piscataway, NJ, USA). Mueller-Hinton (MH) and Brain Heart Infusion agar plates (Thomas Scientific, Swedesboro, NJ, USA) were used as growth media for this purpose. The MIC was determined using a microdilution procedure described in an earlier study [52], where these methods have been described and results presented/discussed.

### 4.3. DNA Plug Preparation and PFGE

DNA plug preparation for PFGE analysis was carried out according to a Laboratory Protocol for molecular typing of *S. aureus* [53]. DNA plugs containing *S. aureus* genomic DNA were then digested with 30 units/plug of restriction enzyme *Sma*I (Promega, Madison, WI, USA) in a 200 μL volume per manufacturer’s recommendations. Digested DNA samples were separated in 1% SeaKem^®^ Gold agarose gels (Cambrex Bio Science Rockland Inc., Rockland, ME, USA). The electrophoresis was carried out for 18.5–19 h in 0.5× TBE (tris-borate- EDTA) using a CHEF Mapper (Bio-Rad, Hercules, CA, USA) at 200 V and 14 °C, with initial and final switch times of 5 and 40 s. BioNumerics software v5.10 (Applied Maths Scientific Software Development, Saint-Martens-Latem, Belgium) was used for cluster analysis. Isolates with a similarity index of ≥ 88% were considered clones. PFGE strain types were compared to the USA100 MRSA isolate [12].

### 4.4. DNA Isolation and Purification

Chromosomal DNA for use in PCR was isolated from *S. aureus* strains grown overnight in a MH broth at 35 °C. Bacterial cell cultures (1 mL) were pelleted by centrifugation at 10,000× *g*, and the pellet was suspended in 180 μL TE buffer (10 mM Tris-HCl + 1 mM EDTA, pH 8.0) containing 10 units of lysostaphin (Sigma, St. Louis, MO, USA). Cell suspension was then incubated at 37 °C for 1 h. This was followed by proteinase K addition and incubation at 55 °C for 1 h. An alkaline lysis (AL) solution (200 μL), made by mixing AL1 and AL2 reagents from a QIAmp kit (Qiagen, Valencia, CA, USA), was added to the tube. Instructions included with the QIAmp kit were followed for *S. aureus* DNA purification. A nanodrop 2000 spectrophotometer (Thermo Fisher Scientific, Wilmington, DE, USA) was used to determine DNA concentrations by reading absorbance at 260 and 280 nm.

### 4.5. Hemolysin Assay

Hemolysin activity was measured using a simple sheep blood agar plate-based assay. Plates containing 7% sheep blood were washed twice with 0.85% sterile NaCl solution. Bacterial strains were then spot inoculated, and plates were incubated at 37 °C for 16–18 h. Plates were then incubated at 4 °C overnight to distinguish hemolysis zones of clearance pertaining to *α, β,* and *δ* hemolysins. Sharp clear edges of the lysis zones indicated the presence of *β* hemolysin. Presence of *α* and *δ* hemolysins was indicated by the observation of discolored zones of clearance without sharp edges as a result of incomplete lysis of hemoglobin [54]. The presence of hemolysin genes *α, β,* and *γ* was also determined by PCR amplification as described earlier [55].

### 4.6. Antimicrobial Resistance Genes

The methicillin resistance determinant gene *mecA* and the heat-stable nuclease gene *nucA* were amplified from *S. aureus* isolates as 533 bp and 270 bp DNA fragments, respectively, using primers and amplification protocols described previously [56]. Antimicrobial resistance genes encoding aminoglycoside resistance (*aac(6′)-aph(2″), ant(4′)-Ia, aph(3′)-IIIa, aac(6′)-Ie-aph(2′)-Ia, str*, and *sat4*) and genes conferring resistance to beta-lactams (*mecA* and *blaZ*), macrolides (*mphC, ermC,* and *ermB*), tetracycline (*tetK* and *tetM*), and chloramphenicol (*catpC221* and *catpC223*) were amplified by using PCR primers and conditions described in previous publications [57,58,59,60,61].

### 4.7. Toxin and Virulence Genes

*S. aureus* chromosomal DNA was used for PCR assays to amplify 19 toxin genes (*sea, seb, sec, sed, see, seg, seh, sei, sej, tst, edin, eta, etb, hla, hlb, hld, hlg, hlg2,* and *PVL*), 12 adhesin genes (*bbp, clfA, clfB, cna, ebpS, fnbA, fnbB, map/eap, sdrC, sdrD, sdrE,* and *spa*), and five other virulence genes (*chp, efb, icaA, V8,* and *arcA*), using primers and conditions described earlier [62,63]. Moreover, all *S. aureus* isolates were also screened for the presence of *arcA* and *agr* groups I to IV genes using PCR primers and conditions described earlier [40]. PCR reactions were performed in a MyCycler thermal cycler (Bio-Rad, Hercules, CA, USA) with HotStart *Taq* polymerase (Qiagen), and PCR products were analyzed by electrophoresis in a 2% agarose gel.

### 4.8. PVL Gene Amplification, MLST and spa Typing

MLST is another discriminatory tool used to characterize bacterial isolates based on the sequencing of ~450-bp internal fragments of seven housekeeping genes [64]. These included acetyl coenzyme A acetyltransferase (*yqiL*), carbamate kinase (*arcC*), glycerol kinase (*glp*), guanylate kinase (*gmk*), phosphate acetyltransferase (*pta*), shikimate dehydrogenase (*aroE*), and triosephosphate isomerase (*tpi*). Primers and amplification procedures for these genes were used as described earlier [65], followed by sequencing of PCR fragments using an ABI prism BigDye Terminator cycle sequencing ready reaction kit (Applied Biosystems, Foster City, CA, USA) and MLSTs mapping via the https://pubmlst.org/organisms/staphylococcus-aureus/ (accessed on 20 April 2021). For each housekeeping gene, the distinct sequences were given different allele numbers. The alleles at each of the seven loci among bacterial isolates represented a series of seven integers and defined its allelic profile. Each allelic profile was assigned a distinct multilocus sequence type or, in short, an ST number (e.g., ST22).

For typing the *spa* region (a 3′ coding region X of the *S. aureus*–specific staphylococcal protein A gene), a polymorphic 24-bp variable-number tandem repeat (VNTR) within this gene was used to discriminate between the outbreak strains of MRSA [66]. The source of variation in this region is due to the duplication or deletion of the 24-bp repetitive units. Therefore, sequencing of the X region of the *spa* gene was carried out with a slight modification of the method described earlier [67] and with an alternate forward primer [68]. Sequencing of the PCR products was done using an ABI prism BigDye Terminator cycle sequencing ready reaction kit (Applied Biosystems, Waltham, MA, USA). *Spa*-types were determined using the website http://spatyper.fortinbras.us/ (accessed on 20 April 2021), following the sequencing of the variable X region of the *spa* gene.

Primers for the *lukS* and *lukF* genes were designed with the primer select module of the Lasergene program (DNASTAR, Inc., Madison, WI, USA) using previously published sequences for these genes (AB186917). Primers were synthesized by Eurofins MWG Operon (Huntsville, AL, USA). Primers for the *lukS* genes were *lukS*-FW 5′-AACAGAAGATACAAGATACAAGTAGCGATAA-3′ and *lukS*-R 5′-GTCTGGCACAAAATAGTCTCT-3′. The primers for *lukF* gene amplification were *lukF*-FW 5′-ACGGTAGGTTATTCTTATG-3′ and *lukF*-R 5′-AATTATTACCTATCCAGTGA-3′. Amplification of other leucotoxin genes such as *lukE-D* (*lukE* + *lukD*) and *lukM* was also carried out using primers and PCR conditions described in a previous publication [69]. Each reaction tube contained 0.1–0.5 µg of bacterial DNA (5 µL), a 10 µM mixture of the forward and reverse primers (5 µL), and 15 µL PCR mix (200 µL PCR mix contained 100 µL 2X DreamTaq PCR Master Mix (ThermoFisher Scientific, Carlsbad, CA, USA), 27 µL of 25 mM magnesium acetate, 66 µL of 10 mM dNTP mix, and 7 µL *Taq* DNA polymerase). An initial denaturation step was carried out at 95 °C for 3 min, followed by 35 cycles consisting of 94 °C denaturation for 60 s, 50 °C annealing for 50 s, and 72 °C extension for 50 s. The extension step in the last cycle was prolonged for 5 min. PCR amplicons were analyzed on 1.5% agarose gels.

### 4.9. SCCmec and agr Group Typing

SCC*mec* types I to V and subtypes IVa, b, c, and d, as described earlier [35,69], were detected with a simple PCR method developed previously [36]. All PCR primers and conditions to amplify these genes were used as described in the publication [36]. Classification of various MRSA isolates into *agr* groups I to IV, as described by Azimian et al. [49], was also performed by PCR using primers and amplification conditions described by Peacock et al. [70].

#### 4.9.1. Determination of the Capsular Polysaccharide Type

A reliable PCR method for the detection of capsular type 5 (361 bp) and for capsular type 8 (173 bp) was used to determine the serotype of all MRSA strains used in the study, by using primers and conditions described earlier [71].

#### 4.9.2. Sequence Analysis

PCR products were purified with a QIAquick gel extraction kit (Qiagen, Germantown, MD, USA), eluted in nuclease-free water, and sequenced by the BigDye sequencing method described above. Nucleotide sequences of the PCR products were BLAST searched against the existing GenBank database to confirm their identity.

## Figures and Tables

**Figure 1 pathogens-10-00918-f001:**
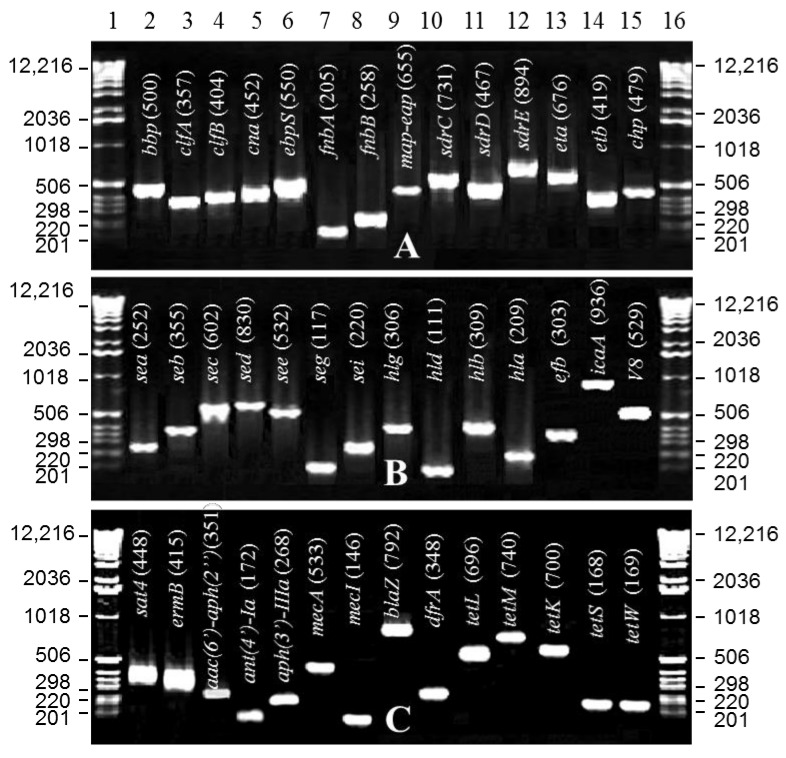
Virulence and antimicrobial resistance gene profile of MRSA isolates. PCR amplicons of various gene products were separated on 2% agarose gels. Lanes 1 and 16, 100-bp DNA ladder; Lanes 2 to 9 and 13, 14, 15 in gel **A** and lanes 2 to 15 in gel **B** indicate the amplicons of different virulence genes from a representative isolate 32. Lanes 10 to 12 in gel **A** represent virulence genes from isolate 50. In gel **C**, lanes 2 to 15 show the amplicons of various antimicrobial resistance genes from a representative isolate 32. PCR amplicon sizes are shown in parentheses.

**Figure 2 pathogens-10-00918-f002:**
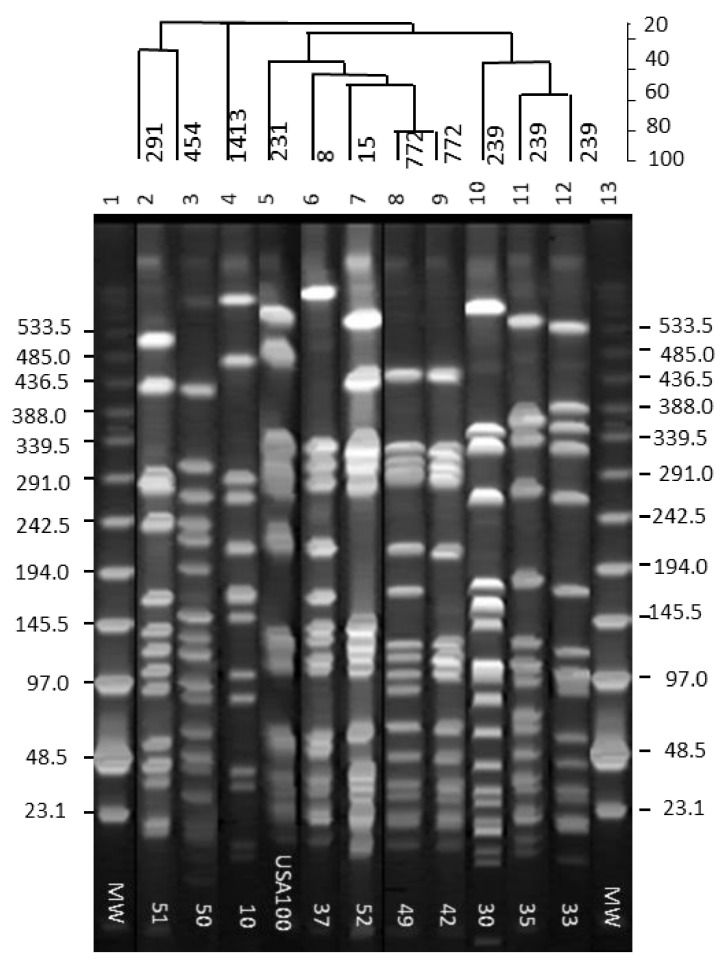
Pulsed-field gel electrophoretic profiles of MRSA isolates in 1% Seakem gold agarose gel. Lane numbers are at the top of the gel, and isolate numbers are shown in the inset. Molecular weight marker sizes are shown on either side of the gel. The % similarity among the isolates is indicated by a dendrogram on top of the gel above the lane numbers.

**Table 1 pathogens-10-00918-t001:** Distribution of antimicrobial resistance genes in MRSA isolates.

Isolate No.	Aminoglycoside	Chloramphenicol	Erythromycin	Tetracycline	Trimethoprim	Methicillin	Beta-Lactam
10	[*aac(6′)-Ie-aph(2′)-Ia, str*] [*aac(6′)-Ie-aph(2)-Ia, str*]	[*catpC221* and *catpC223*]	[*ermB, sat4*]	[*tetL tetM, tetS, tetW*]	[*dfrA*]	[*mecA*]	[*ND*]

15	[*aac(6′)-aph(2″),*	*[cat(pC194), cat(pC221), catpC223]*	[*ND*]	[*tetM, tetS, tetW*]	[*dfrA*]	[*mecA*]	[*ND*]
25	[*aac(6′)-aph(2″), aph(3′)-IIIa,* *aac(6′)-Ie-aph(2’)-Ia, str*]	[*cat(pC194), cat(pC221), catpC223*]	[*sat4*]	[*tetL, tetM, tetS, tetW*]	[*dfrA*]	[*mecA*]	[*ND*]

30	[*aph(3′)-IIIa, aac(6′)-Ie-aph(2′)-Ia*]	[*cat(pC194), cat(pC221), catpC223*]	[*sat4*]	[*tetM, tetS, tetW*]	[*dfrA*]	[*mecA*]	[*ND*]
31	[*aac(6′)-aph(2″), aac(6′)-Ie-aph(2′)-Ia*]	[*cat(pC194), cat(pC221), catpC223*]	[*ermB*]	[*tetM, tetS, tetW*]	[*dfrA*]	[*mecA*]	[*blaZ*]
	[*aac(6′)-aph(2″), aph(3′)-IIIa,* *aac(6′)-Ie-aph(2′)-Ia, str*]	[*cat(pC194), cat(pC221), catpC223*]	[*ermB*]	[*tetM, tetS, tetW*]	[*dfrA*]	[*mecA*]	[blaZ]

37	[*aac(6′)-aph(2″), aph(3′)-IIIa*	[*cat(pC194), cat(pC221), catpC223*]	[*ermB, sat4*]	[*tetM, tetS, tetW*]	[*dfrA*]	[*mecA*]	[*ND*]
40	[*aac(6′)-aph(2″), ant(4′)-Ia, aph(3′)-IIIa*, *aac(6′)-Ie-aph(2′)-Ia, str*]	[*cat(pC194), cat(pC221), catpC223*]	[*ermB, sat4*]	[*tetM, tetS, tetW*]	[*dfrA*]	[*mecA*]	[*blaZ*]

42	[*aph(3′)-IIIa, aac(6′)-Ie-aph(2′)-Ia, str*]	[*cat(pC194), cat(pC221), catpC223*]	[*ermB]*	[*tetM, tetS, tetW*]	[*dfrA*]	[*mecA*]	[*blaZ*]
49	[*aac(6′)-aph(2″), ant(4′)-Ia*, *aac(6′)-Ie-aph(2′)-Ia, str*]	[*cat(pC194), cat(pC221), catpC223*]	[*ermB, sat4*]	[*tetM, tetS, tetW*]	[*dfrA*]	[*mecA*]	[*blaZ*]

32	[*aac(6′)-aph(2″), ant(4′)-Ia, aph(3′)-IIIa*, *aac(6′)-Ie-aph(2′)-Ia, str*]	[*cat(pC194), cat(pC221), catpC223*]	[*ermB, sat4*]	[tetK, tetL, *tetM, tetS, tetW*]	[*dfrA*]	[*mecA*]	[*blaZ*]

33	[*aac(6′)-aph(2″), aac(6′)-Ie-aph(2′)-Ia, str*]	[*cat(pC194), cat(pC221), catpC223*]	[*ermB*]	[tetK, *tetM, tetS, tetW*]	[*dfrA*]	[*mecA*]	[*blaZ*]
34	[*aac(6′)-aph(2″), aac(6′)-Ie-aph(2′)-Ia, str*]	[*cat(pC194), cat(pC221), catpC223*]	[*ermB*]	[*tetM, tetS, tetW*]	[*dfrA*]	[*mecA*]	[*blaZ*]
38	[*aac(6′)-aph(2″), aph(3′)-IIIa*, *aac(6′)-Ie-aph(2′)-Ia, str*]	[*cat(pC194), cat(pC221), catpC223*]	[*ermB, sat4*]	[*tetM, tetS, tetW*]	[*dfrA*]	[*mecA*]	[*blaZ*]

41	[*aac(6′)-aph(2″), aac(6′)-Ie-aph(2′)-Ia, str*]	[*cat(pC194), cat(pC221), catpC223*]	[*ermB*]	[*tetM, tetS, tetW*]	[*dfrA*]	[*mecA*]	[*blaZ*]
48	[*aac(6′)-aph(2″), aac(6′)-Ie-aph(2′)-Ia, str*]	[*cat(pC194), cat(pC221), catpC223*]	[*ermB*]	[*tetM, tetS, tetW*]	[*dfrA*]	[*mecA*]	[*blaZ*]
50	[*aac(6′)-Ie-aph(2′)-Ia, str*]	[*ND*]	[*sat4*]	[*tetS, tetW*]	[*dfrA*]	[*mecA*]	[*blaZ*]
51	[*ant(4′)-Ia*, and *aph(3′)-IIIa*, *aac(6′)-Ie-aph(2′)-Ia, str*]	[*cat(pC194), cat(pC221), catpC223*]	[*ermB, sat4*]	[*tetK, tetS, tetW*]	[*dfrA*]	[*mecA*]	[*blaZ*]

52	[*ant(4′)-Ia*, and *aph(3′)-IIIa*, *aac(6′)-Ie-aph(2′)-Ia*]	[*cat(pC194), cat(pC221), catpC223*]	[*ermB, sat4*]	[*tetS, tetW*]	[*dfrA*]	[*mecA*]	*blaZ*]


ND = Not detected.

**Table 2 pathogens-10-00918-t002:** MLST, *spa*, SCC*mec*, *agr*, and capsular typing of MRSA isolates.

PFGE Profile	Isolate No	Origin	Infection Type	Symptoms/Reasons Admitted for	MLST Type	*spa* Type	SCC*mec* Type	*agr* Group	Capsular Genotype
1	32	P	HA	DNR Colitis Tracheostomy	ST239	t030	III	I	8
1	33 *	P	CA	Septic Shock	ST239	t030	III	I	8
1	34	P	CA	Intestinal Obstruction	ST239	t030	III	I	NT
1	41	P	HA	Anxiolytic Poisoning	ST239	t030	III	I	8
1	48	P	CA	SOB, Fever	ST239	t030	III	I	8
1	38	P	HA	CuSo4 Poisoning	ST8	New	V	I	5
2	15	N	HA	Hepato-Splenomegaly	ST239	New	III	I	8
2	25 *	N	CA	Chronic Renal Failure	ST8	t064	IVa	I	5
2	31 *	N	HA	Tetanus, Locked Jaw	ST239	t030	III, IVa	I	8
2	37	N	HA	Cardiogenic shock	ST8	t064	IVa	I	5
2	40 *	N	HA	Kyphoscoliosis	ST30	t138	IVa	I	8
3	10 *	N	HA	Guillain Barr Syndrome	ST1413	t314	III, V	I	8
4	30	N	HA	Acute unconsciousness	ST239	t987	III	I	8
5	35	N	HA	Dilated Cardiomyopathy	ST239	t030	III	I	8
6	42	N	CA	Hepato-Splenomegaly	ST772	t5414	III	I	5
7	49	N	CA	Intestinal Obstruction	ST772	t5414	III	I	5
8	50	P	HA	CuSo4 Poisoning	ST503	t138	III, IVa, V	I	5
9	51 *	N	HA	Chronic Renal Failure	ST291	t1149	III, V	I	NT
10	52	P	HA	Neuropathy, Tuberculosis	ST15	t1509	III, IVa, V	I	5

P = Perirectal; N = Nasal; CA = Community-acquired; HA = Hospital-acquired; NT = Non-typable; * Patient died.

**Table 3 pathogens-10-00918-t003:** Differences in the *spa*-types of various MRSA isolates.

X-Region of the Staphylococcal Protein A Gene		*spa* Type	Isolate #
5′ FL-**W1 G1 K1 A1 Q1 K1 A1 O1 M1 Q1**-3 FL′		t987	30
5′ FL-**Y1 H1 G1 C1 M1 B1 Q1 B1 L1 O1** AAGAAGATGGTAACGGAGTACATG-3′ FL		t064	25, 37
5′ FL-**T1 K1 J1 E1 F1 M1 B1 P1 B1**-3′ FL		t5414	42, 49
5′ FL-**W1 G1 K1 A1 Q1 Q1**-3′ FL		t030	31-35, 41, 48
5′ FL-**X1 K1 A1 O1 M1 Q1**-3′ FL		t138	40, 50
5′ FL-**X1 M1 J1 H2 M1**-3′ FL		t314	10
5′ FL-**U1 J1 G1 J1**-3′ FL		t1509	52
5′ FL-**Y1 H1 G1 C1 M1 B1 Q1 B1 L1 A**AATAAGATGGTAACTGATTACATG-3′ FL		New	38
5′- *CCCAAGACAGCAACAAGCCTGGTA* **A**GAGAGGACGGCAACAAACCTGGT-		New	15
**A**AAGAAGACAGCAAAAAAACTGGCA*AGACGATGGCAACAAGCCGGGCA*--3′ FL			
5′- *TGNAAGACGGCAACAAACCTGGTA* **A**AAGAAGACAACGAAAAACCTGGT		New	51
**A**AAGAAGATGGCAACTAGCCTGGT **B1 B1** AAAGAAGACGGCTACAAGCCTGGT--3′ FL			
**A1**: AAAGAAGACAACAAAAAACCTGGC	**B1**: AAAGAAGACAACAAAAAACCTGGT	**C1:** AAAGAAGACAACAAAAAGCCTGGC	
**E1**: AAAGAAGACGGCAACAAACCTGGC	**F1**: AAAGAAGACAACAACAAGCCTGGC	**G1:** AAAGAAGACAACAACAAGCCTGGT	
**H1**: AAAGAAGACAATAACAAGCCTGGC	**H2**: AAAGAAGATGGCAACAAGCCTAGT	**J1**: AAAGAAGACGGCAACAAACCTGGT	
**K1:** AAAGAAGACGGCAACAAACCTGGT	**L1**: AAAGAAGACGGCAACAAGCCTGGC	**M1**: AAAGAAGACGGCAACAAGCCTGGT	
**O1**: AAAGAAGATGGCAACAAACCTGGT	**P1**: AAAGAAGATGGCAACAAGCCTGGC	**Q1**: AAAGAAGATGGCAACAAGCCTGGT	
**T1:** GAGGAAGACAACAAAAAACCTGGT	**X1**: GAGGAAGACAACAACAAGCCTGGT	**Y1**: GAGGAAGACAATAACAAGCCTGGC	
**W1**: GAGGAAGACAACAACAAGCCTGGC	**5′-FL:** TAAACGATGCTCAAGCACCAAAAG	**3′-FL**: AAGAAGACGGCAACGGAGTACATG	

Bold and numbered letters indicate a single letter code for polymorphic 24-bp VNTR sequences within the 3′ coding region X of the spa gene.New VNTRs are shown with their full length 24-bp sequences indicated by a bolded starting letter. The 5′ and 3′ flanking regions containing variations from the indicated sequences are italicized and underlined.

**Table 4 pathogens-10-00918-t004:** Distribution of Staphylococcal enterotoxin, adhesion, hemolysin, and other virulence genes in human clinical MRSA isolates.

Isolate No.	Toxin Genes	Adhesin Genes	Other Virulence Genes	Hemolysin Activity
10	[*eta, sea, seb, sec, sei, sej, **hla, hlb, hld, hlg, pvl** (**lukS, lukF, lukM, lukE-D**), **sed, see, seg, seh, tst***]	(*ebpS, **bbp, can, clfA, clfB, fnbA, fnbB, map-eap, spa***)	(*ica, **cfb**, v8*)	*γ*
15	[*seb, sec, **hla, hlb, hld, hlg, pvl (lukE-D), sed, see, seg, seh, tst***]	(*ebpS*, ***bbp, can, clfA, clfB, fnbA, fnbB, map-eap, spa***)	(***cfb**, v8*)	*α*
25	[*sea, seb, sec, **hla, hlb, hld, hlg, pvl** (**lukE-D**), **sed, see, seg, seh, tst***	(***bbp, can, clfA, clfB, fnbA, fnbB, map-eap, spa***)	(*ica, **cfb**, v8*)	*α*
30	[*eta, sea, seb, sec, sei, sej**, hla, hlb, hld, hlg, pvl** (**lukS, lukF, lukM, lukE-D**), **sed, see, seg, seh, tst***]	(***bbp**, **can, clfA, clfB, fnbA, fnbB, map-eap, spa***)	(*chp, ica, **cfb**, v8*)	*γ*
31	[*eta, seb, sec, sei, **hla, hlb, hld, hlg, pvl** (**lukE-D**), **sed, see, seg, seh, tst***]	(*ebpS, **bbp, can, clfA, clfB, fnbA, fnbB, map-eap, spa***)	(*ica, **cfb**, v8*)	*γ*
35	[*eta, seb, sec, sei, sej, **hla, hlb, hld, hlg, pvl** (**lukS, lukF, lukM, lukE-D), sed, see, seg, seh, tst***]	(***bbp, can, clfA, clfB, fnbA, fnbB, map-eap, spa***)	(*chp, ica, **cfb**, v8*)	*γ*
37	[*eta, sea, seb, sec, sej, **hla, hlb, hld, hlg, pvl** (**lukE-D**), **sed, see, seg, seh, tst***]	(***bbp, can, clfA, clfB, fnbA, fnbB, map-eap, spa***)	(*chp, ica, **cfb**, v8*)	*α*
40	[*eta, sea, seb, sec, sej, **hla, hlb, hld, hlg, pvl** (**lukS, lukF, lukM, lukE-D**), **sed, see, seg, seh, tst***]	(***bbp, can, clfA, clfB, fnbA, fnbB, map-eap, spa***)	(*chp, ica, **cfb**, v8*)	*β*
42	[*eta, seb, sec, sei, **hla, hlb, hld, hlg, pvl** (**lukS, lukF, lukM, lukE-D), sed, see, seg, seh, tst***]	(*ebpS, **bbp, can, clfA, clfB, fnbA, fnbB, map-eap, spa***)	(*chp, ica, **cfb**, v8*)	*α*
49	[*eta, sea, seb, sec, sei, sej, **hla, hlb, hld, hlg, pvl** (**lukS, lukF, lukM, lukE-D), sed, see, seg, seh, tst***]	(***bbp, can, clfA, clfB, fnbA, fnbB, map-eap, spa***)	(*chp, ica, **cfb**, v8*)	*β*
32	[*eta, sea, seb, sec, sei, **hla, hlb, hld, hlg, pvl** (**lukS, lukF, lukM, lukE-D), sed, see, seg, seh, tst***]	(*ebpS, **bbp, can, clfA, clfB, fnbA, fnbB, map-eap, spa***)	(*chp, ica, **cfb**, v8*)	*γ*
33	[*eta, sei, sej**, hla, hlb, hld, hlg, pvl** (**lukS, lukF, lukM, lukE-D), sed, see, seg, seh, tst***]	(*ebpS, **bbp, can, clfA, clfB, fnbA, fnbB, map-eap, spa***)	(*chp, ica, **cfb**, v8*)	*γ*
34	[*eta, sec, sei, **hla, hlb, hld, hlg, pvl** (**lukS, lukF, lukM, lukE-D), sed, see, seg, seh, tst*****]**	(*ebpS, **bbp, can, clfA, clfB, fnbA, fnbB, map-eap, spa***)	(*chp, ica, **cfb**, v8*)	*γ*
38	[*eta, sea, seb, sec, sei**, hla, hlb, hld, hlg, pvl** (**lukS, lukF, lukM, lukE-D**), **sed, see, seg, seh, tst***]	(***bbp, can, clfA, clfB, fnbA, fnbB, map-eap, spa***)	(*chp, ica, **cfb**, v8*)	*β*
41	[*eta, seb, sec, sei, sej, **hla, hlb, hld, hlg, pvl** (**lukS, lukF, lukM, lukE-D**), **sed, see, seg, seh, tst***]	(*ebpS, **bbp, can, clfA, clfB, fnbA, fnbB, map-eap, spa***)	(*chp, ica, **cfb**, v8*)	*α*
48	[*eta, seb, sec, sei, sej, **hla, hlb, hld, hlg, pvl** (**lukS, lukF, lukM, lukE-D**), **sed, see, seg, seh, tst*****]**	(*ebpS, **bbp, can, clfA, clfB, fnbA, fnbB, map-eap, spa***)	(*chp, ica, **cfb**, v8*)	*α*
50	[*sea, seb, sec, sei, sej**, hla, hlb, hld, hlg, pvl** (**lukS, lukF, lukM, lukE-D**), **sed, see, seg, seh, tst*****]**	(***bbp, can, clfA, clfB, fnbA, fnbB, map-eap, spa***)	(*chp, ica, **cfb**, v8*)	*β*
51	[*eta, sea, seb, sec**, hla, hlb, hld, hlg, pvl** (**lukS, lukF, lukM, lukE-D**), **sed, see, seg, seh, tst*****]**	(*ebpS, **bbp, can, clfA, clfB, fnbA, fnbB, map-eap, spa***)	(*chp, ica, **cfb**, v8*)	*β*
52	[*eta, sea, seb, sec**, hla, hlb, hld, hlg, pvl** (**lukS, lukF, lukM, lukE-D**), **sed, see, seg, seh, tst***]	(*ebpS, **bbp, can, clfA, clfB, fnbA, fnbB, map-eap, spa***)	(*chp, ica, **cfb**, v8*)	*α*

All the genes that are in bold and italic were common among all the isolates.

## Data Availability

Not applicable.

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
