# Peer review of "Genotypic Characterization of Clinical Isolates of Staphylococcus aureus from Pakistan"

_pathogens, 2021, doi:10.3390/pathogens10080918_

Round 1
Reviewer 1 Report
Comments to authors:
In my opinion, the article is well written and the study was properly designed, with exhaustive introduction and discussion section. Probably the authors have to re-organize tables and figures.
Line 11: correct after “Pakistan”.
Table 1 and the following tables: the font is too small, difficult to read. For Table 1 please define “NA” for isolate 50.
Lines 110-113: figure 2 is not clear in relation to the text; in particular, there are only the isolates 10, 30, 35, 42, 49 (not 32, 33, 34 etc.); explain better.
Line 158 and line 160: Figure 2 represents PFGE, not virulence genes.
Line 172: it is difficult to distinguish between bold and italic font in table 4; correct and use higher font.
Lines 180-183: not clear sentence.
Line 237: correct that.
Line 329: define ICU.
Line 435: correct the bracket.
Author Response
We appreciate a critical review of the manuscript by the reviewer to improve the quality of the manuscript. Please find attached the response to reviewer's comments.

Reviewer 2 Report
The purpose of the research presented in the manuscript was genetic analysis of MRSA strains isolated from patients in Pakistan.
The authors used a number of genetic methods for their characterization and a few phenotypic methods. Although many results were obtained, they were presented in a disorderly manner.
The reviewer noticed the ambiguities in each part of manuscript. The chapter: “Material and method”:• There is a lack of information on the number of tested strains. What were the criteria for selecting strains for testing? (in section 4.1)• There is a lack of information whether each strain derived from one patient or not? (in section 4.1)• While the Authors described the disc method, there is no information about the dilution method. For which antibiotics the MIC was determined? (in section 4.2)• Why the Authors used 3 methods to determine drug resistance in bacteria?• There are no references in “DNA isolation and purification” (in section 4.4).
- What about control strains for PCR assays to amplify toxin and adhesion genes? (in section 4,7)
- The Authors state that they conducted PVL typing. This is not correct. Authors investigated the presence of genes for various leukocidins, including PVL (in section 4.8)
- The sentence “All PCR primers and conditions to amplify these genes were used as described in the publication” requires a reference to the literature (in section 4.9)
The chapter: “Results”:• There is no information about the results of the MIC tests (in section 2.1). The table 2 is not clear. • Table 1 is incomplete. The resistance profile to antibiotics obtained by the disc diffusion method is missing. There is also a lack of MIC values and reference to the presence of resistance genes. The table with the antibiotic resistance profile obtained in the disk method would be more readable. • Figure 1 is difficult to read. The genetic profiling of one strain requires searching for different pathways on three gels• Table 2 (in section 2.3) shows the results of genetic typing of MRSA strains using several methods (MLST, spa, SCC mec, agr and capsular typing), while the title of section 2.3 shows something else (not capsular typing, but PVL typing). The results of the different leukocidin gene assays are not shown in the table 2 and in all section 2.3. • It is unclear on what basis the Authors of the manuscript identified the infectious type as HA or CA in table 2.• The data in Table 3 are not informative, it can be removed• Table 4 is not transparent. The Authors used small print and lots of data.
The Authors should list the genes present in each strain in the text, and include only the remaining virulence genes results in the table. • The extensive discussion contains too many results and too few interpretations by the Authors. The Authors did not provide specific conclusions from the research. The discussion requires deeper reflections, because it is difficult to know e. g. which MRSA clones dominate in Pakistan in terms of pathogenic potential and drug resistance. This is what the reader of this publication expects.• Statement: “The presence of multidrug-resistant MRSA isolates in Pakistan indicated poor hygienic condition…” (Abstract section) is not entirely true. The presence of MDRSA is indicative an overuse of antibiotics and a lack of rational antibiotic therapy.
Author Response

(The authors gave the same response as above.)

Round 2
Reviewer 2 Report
Most of the reviewer's comments were taken into account by the authors of the manuscript.
Some comments still need to be completed
- there is a lack of information whether each strain derived from one patient or not? (in section 4.1).
- information about the eligibility criteria for HA-MRSA or CA-MRSA should be included in the manuscript. If the authors do not have it, the HA and CA markings should be removed.
Author Response
Comment 1. There is a lack of information whether each strain derived from one patient or not? (in section 4.1).
Response: It is mentioned on line 334 that the strains were obtained from multiple patients. An extra line will be added stating that each strain came from an individual patient.
Comment 2. Eligibility criteria for HA-MRSA or CA-MRSA should be included in the manuscript.
Response: Added the following to section 4.1. HA-MRSA isolates represent the isolates obtained from patients that acquired MRSA while undergoing treatment in hospital and CA-MRSA isolates represents those that were isolated from patients that came with MRSA infections and admitted to the hospital.